# Acquisition and Processing of Wider Bandwidth Seismic Data in Crystalline Crust: Progress with the Metal Earth Project

**Mostafa Naghizadeh [1],\*****, David Snyder [1], Saeid Cheraghi [1], Steven Foster [2], Samo Cilensek [2], Elvis Floreani [2] and Jeff Mackie [3]**

[1] Harquail School of Earth Sciences, Laurentian University, Sudbury, ON P3E2C6, Canada; dbsnyder1867@gmail.com (D.S.); SCheraghi@laurentian.ca (S.C.)

[2] Absolute Imaging Inc., Calgary, AB T2R1J2, Canada; steve@absoluteimaging.ca (S.F.); samo@absoluteimaging.ca (S.C.); elvis@absoluteimaging.ca (E.F.)

[3] SAExploration Ltd., Calgary, AB T2B3M2, Canada; jmackie@saexploration.com

\* Correspondence: mnaghizadeh@laurentian.ca

**Abstract:** The Metal Earth project acquired 927 km of deep seismic reflection profiles from August to November of 2017. Seismic data acquired in this early stage of the Metal Earth project benefited greatly from recent advances in the petroleum sector as well as those in mineral exploration. Vibroseis acquisition with receivers having a 5 Hz response (10 dB down) generated records from a sweep signal starting at 2 Hz, sweeping up to 150 Hz or 200 Hz. Not only does this broadband signal enhance reflections from the deepest to the shallowest crust, but it also helps the use of full waveform inversion (e.g., to mitigate cycle-skipping) and related techniques. Metal Earth regional-scale transects using over 5000 active sensors target mineralizing fluid pathways throughout the crust, whereas higher spatial-resolution reflection and full-waveform surveys target structures at mine camp scales. Because Metal Earth was proposed to map and compare entire Archean ore and geologically similar non-ore systems, regional sections cover the entire crust to the Moho in the Abitibi and Wabigoon greenstone belts of the Superior craton in central Canada. Where the new sections overlap with previous Lithoprobe surveys, a clear improvement in reflector detection and definition is observed. Improvements are here attributed to the increased bandwidth of the signal, better estimates of refraction and reflection velocities used in processing, and especially the pre-stack time migration of the data.

**Keywords:** mineral exploration; seismic reflection methods; hard rock exploration; archean ore systems

## 1. Introduction

Innovative technology for conducting seismic exploration historically derives from petroleum exploration in sedimentary (soft) rock environments, but mineral exploration in crystalline (hard) rock environments requires different emphasis [1]. Major innovations and more incremental technical improvements have occurred simultaneously within the petroleum seismic exploration industry over the past decade and these were adapted into the Metal Earth seismic acquisition program [2]. Metal Earth is dedicated to understanding the processes responsible for the differential metal endowment in Archean greenstone provinces and to do so will, for the first time, map entire ore and non-ore systems at full crust-mantle scale to identify key geological-geochemical-geophysical attributes of metal sources, transport pathways, and economic concentrations. Metal Earth, therefore, requires new observations and data over a broad range of scales, from craton- to deposit-scale and integration of information from seismic, magnetotelluric (MT), gravity and traditional geological mapping surveys. The primary mode

of transect surveying, deep seismic reflection profiling, will build upon previous regional-scale surveys conducted as part of the Lithoprobe and Discover Abitibi projects [3–6]. Acquisition technology is significantly improved from the early days of Lithoprobe (1990), less so from the time of Discover Abitibi (2005). A few seemingly small improvements however now enable very significant new approaches to analysis and enhance our understanding of mineralization pathways and processes. Here the focus will be solely on the advances adopted in the seismic reflection profiling method, particularly those embracing broader bandwidth data and its migration. In this article, we will only discuss the seismic processing workflow for the crustal scale (R1) seismic data. The processing and analysis of high-resolution (R2) and full waveform inversion tailored (R3) seismic data is currently underway and will be discussed in future publications.

## 2. Seismic Data Acquisition

### 2.1. Improved Bandwidth

Perhaps the most straightforward improvement is in the bandwidth of seismic signal now recorded. It has long been known that frequencies higher than 50–70 Hz do not propagate well into the deep crust whereas low frequency propagation appears relatively unlimited [7]. The natural sources such as earthquakes are recorded across all the continents at less than 10 Hz if sufficiently large in magnitude. Local or regional seismic surveys using so-called controlled sources typically were limited to frequencies greater than 10 Hz by the practicalities of the technology (e.g., limitation of the Vibroseis sources being able to start production sweep only at 6–8 Hz) and the large amount of equipment typically required to acquire multi-fold seismic data. That has changed recently.

The newest large (61,800 lbs of peak force) vibrator trucks that are typically used as precisely controlled seismic sources by the petroleum industry now attain a theoretical bandwidth of 2 Hz to 250 Hz (Figure 1). The AHV-IV 364 Commanders contracted by Metal Earth from SAExploration (SAExploration Ltd., Calgary, AB, Canada) attain a peak force at 6.2 Hz, but useful frequencies have been recorded as low as 3 Hz (Figure 2). A similar Vibroseis system was used to acquire the PolandSPAN regional seismic survey (2200 km) using a custom broadband sweep of 2–150 Hz in 2011 [8]. Metal Earth used an array of four of these vibrator trucks producing a linear upsweep of 2–96 Hz that was repeated four times at each nominal source location. Based on the relatively fast seismic wave speeds previously encountered near the surface in these greenstone belts, useful P-wave seismic wavelengths of 50–3000 m were thus generated. In a high-resolution acquisition mode (Table 1), an upsweep of 5–120 Hz was repeated four times but with a 6.25 m move up between each sweep and these results will be described elsewhere. Strong, near-source S-wave conversions similar to those of some Discover Abitibi transects are again observed at the alluvium-basement discontinuity several meters below the surface [9].

Receiver bandwidth has also improved. Metal Earth deployed single SG-5 vertical-component 5-Hz geophones within an OYO GSX Wireless nodal recording system. Although these geophones have a 5-Hz natural frequency, the recorded signal is down only a few dB at 3 Hz (Figure 2). In the cable-less recording system used, each sensor with its associated recording box and power pack were harvested when that sensor location was no longer required within the specified symmetrical receiver array. Lithoprobe-standard split-spread receiver arrays with 15-km far offsets were used for longer transects. Regional mode surveying used 50 m source and 25 m receiver intervals. For shorter transects, the entire spread remained fixed and active throughout the shooting. Long offsets continue to be valuable for deep velocity analysis and in order to capture reflections off steeply dipping in-line structures. Wireless receiver spacing of 12.5 m was used in a second acquisition mode, so-called high-resolution 2-D transects, and single sweeps at 6.25 m shot spacing makes the nominal common depth point (CDP) intervals of 3 m available if needed for better resolution. With frequencies of 120 Hz recorded to a few kilometers depth, vertical resolution of several tens of meters is theoretically possible.

Recent 3-D surveys in mine camps using explosive sources have recorded useful 80–170 Hz signal that enables the eventual correlation with rock units with 5–10 m resolution [10].

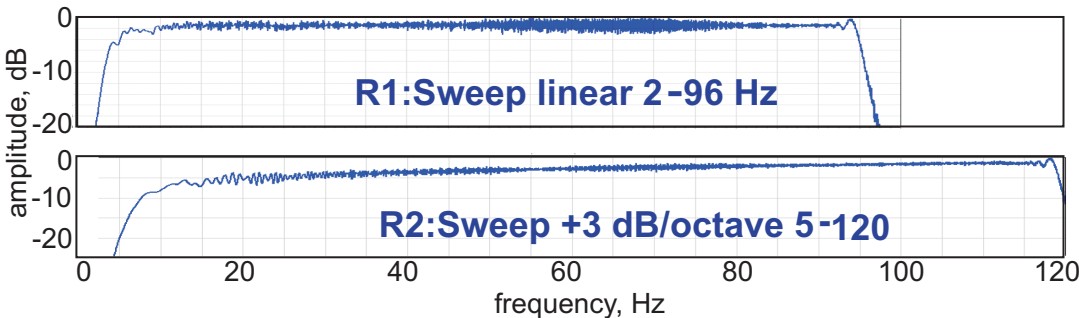

**Figure 1.** Observed frequency content of the vibrator sweeps (weighted sum ground force) as recorded on the baseplates during source quality control: (**Top**) regional (R1) mode, (**Bottom**) high-resolution (R2) mode.

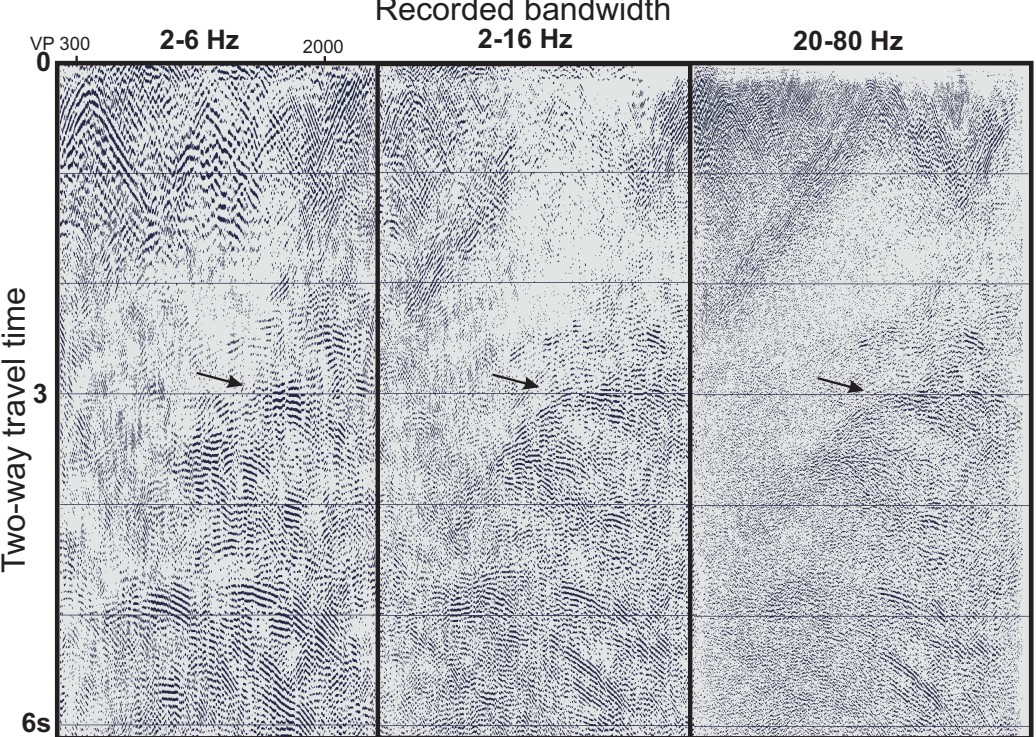

**Figure 2.** Recorded seismic frequencies as illustrated by band-pass sections. The ranges on top of the plots indicate the band-pass frequencies allowed. Note that high-amplitude reflections recorded in traditional band-pass ranges are also observed at frequencies less than 6 Hz.

**Table 1.** Acquisition parameters used in regional, high-resolution, and full-waveform modes.

| Parameter | Regional (R1) Mode | High-Resolution (R2) Mode | Full-Waveform (R3) Mode |
|---|---|---|---|
| Record length | 12 or 16 s | 12 s | 12 s |
| Sample rate | 2 ms | 2 ms | 2 ms |
| Spread size | 15 km–0–15 km | All live (10–20 km) | All live (30–80 km) |
| Roll on/off | Yes | Yes | Yes |
| Source interval | 50 m (4 sweeps); 12.5 m (1 sweep) | 25 m (4 sweeps); 6.25 m (1 sweep) | As in R1 |
| Receiver interval | 25 m | 12.5 m | 25 m |
| Vibrator sweep | 28 s, 2–96 Hz linear; 4 vibs; | 28 s, 5–120 Hz + 3db/octave; 3 vibs; | As in R1 |

## 2.2. Source Arrays

Source arrays are designed to help reduce the strength of surface waves and other noise generated by the large trucks used as the seismic source. Array design involves the number of vibrator trucks used, the number of sweeps to be added together as a single effective source, and the distance between the individual vibrator trucks while vibrating. Metal Earth seismic surveys were acquired with 12.5 m and 6.25 shots spacing between each single sweep per station for R1 and R2 acquisition scenarios, respectively (Table 1). It was anticipated that vibrator points (VPs) could be grouped to form larger arrays during the processing if a stronger source signal was required. For example, four neighbouring sweeps could be grouped to form 50 m source intervals. This large spacing between individual sweeps was adapted in order to make possible a higher spatial resolution, but also to mitigate damage to paved roads arising from repeated sweeps in the same location. Three early seismic transects used source arrays similar to earlier Lithoprobe and Discover Abitibi surveys. The Chibougamau transect was acquired entirely on gravel roads and used four sweeps in one location. The Malartic and the Rouyn-Noranda transects used 1 m move up between each of four sweeps. The use of four vibrators for every R1 sweep effectively made an 38-m-long source array and thus mitigated some of the horizontally propagating seismic noise (Figure 3). Testing at the beginning of the seismic program, on the Chibougamau transect, showed little difference between the 0 m and 6.25 m move up between each sweep in stacked sections using 50 m source intervals.

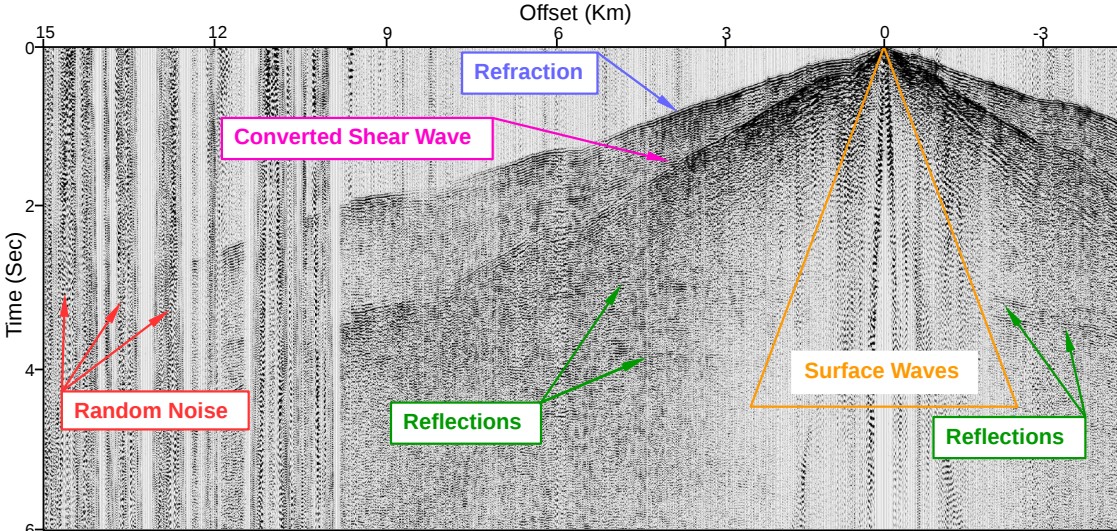

**Figure 3.** A shot gather from Metal Earth's Malartic R1 Survey.

## 2.3. Toward Using Full Waveforms

In addition to operating at regional and high-resolution modes to create traditional CDP gathers and stacks, some transects were augmented so as to be analyzed using the full recorded seismic waveforms. More commonly referred to as full waveform inversion (FWI) [11,12], this method requires a receiver array as long and as dense as logistics and budget will allow. Metal Earth acquired data

suitable for full waveform inversion along three 2-D lines, using a 40–80-km long linear receiver array with both vibrator and explosive sources all recorded by the entire active array (Figure 4; Table 2). The Sudbury transect consists of a grid of short (10–15 km long) lines designed for full waveform inversion as well. The vibrator source locations were the same as the regional (R1) locations along the receiver spread; shots were spaced at about 10 km intervals along the R3 profile. The shots were intended to provide the 1–5 Hz frequencies required for the first few inversion iterations, supplemented by the higher-frequency and spatially denser vibrator sources during subsequent iterations.

To our knowledge, FWI data and analysis has not been used at this scale in hard rock environments to date although related, more traditional near-surface P- and S-wave tomography has shown promising results [9]. The FWI is performed in stages using narrow bands of data—both in time and frequency domains [12,13]. Starting frequencies of 1 Hz or 2 Hz are typical in soft-rock settings. High computational costs typically limited inversions to a few stages that reach maximum frequencies of perhaps 10 Hz. These frequencies translate into seismic wavelengths of 500–5000 m with mapping resolution typically reaching the theoretical limit of $\frac{1}{2}$ wavelength. Additional inversion stages at higher frequencies would greatly enhance the resolution achievable in modeled P-wave sections but are computationally expensive. The primary strength of this FWI method is that it maps P-wave velocity structure at equal resolution vertically and horizontally so that nearly vertical structures will be revealed as clearly as horizontal ones wherever sufficiently distinct changes in rock types exist. Such changes may be related to lithology or degree of alteration/mineralization. By using offsets as great as 80 km, we hope to map structures as deep as 10–12 km and be able to undershoot problem logistical areas such as swamps, mines or towns (Figure 4). At shallower depths of a few kilometers, both the high-resolution CDP and FWI modes are readily mapped against drill core compilations of known rock to extrapolate these know rocks over greater rock volumes in mine camps [10]. The crooked seismic lines of the Metal Earth project can also benefit from 2.5D FWI analysis with the potential of generating higher resolution subsurface velocity models [14,15].

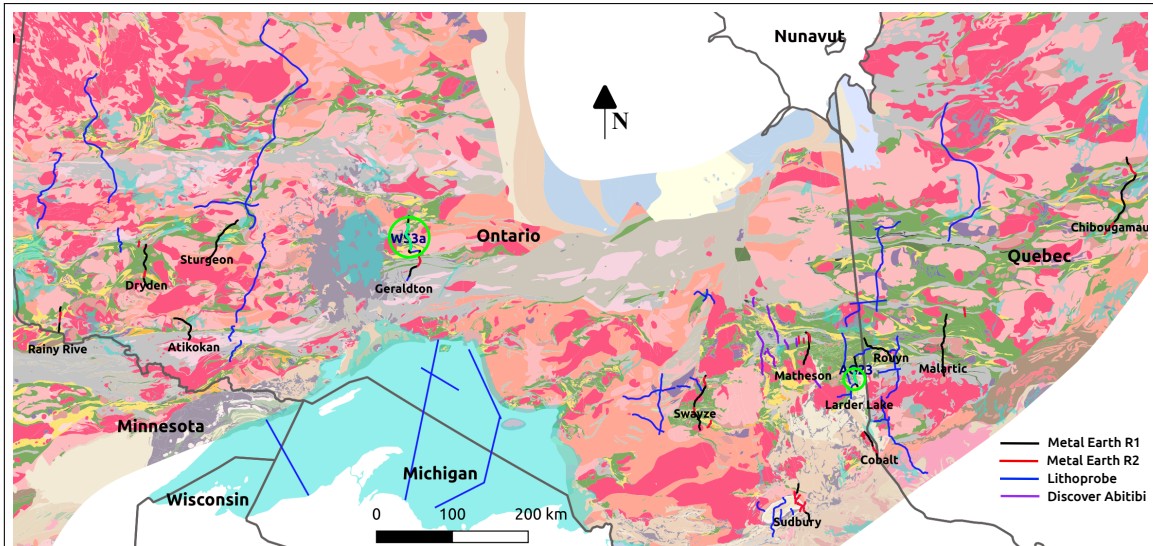

**Figure 4.** Location of the thirteen new Metal Earth transects, compared with previous Lithoprobe and Discover Abitibi profiles. The green lines, highlighted also by green circles, show the location of overlapped parts of Metal Earth and Lithoprobe transects.

**Table 2.** Individual transects of Metal Earth, including both regional and high-resolution modes.

| Transect Name | Length, km | Acquisition Modes | Comment |
|---|---|---|---|
| Chibougamau | 129.85 | R1, R2 × 2 | No vibrator move-up |
| Malartic | 84.775 | R1, R3 | Major gap near Malartic Mine/Town |
| Rouyn-Noranda | 84.775 | R1, R3 | Crooked line; coincident with Lithoprobe AG-21 |
| Larder Lake | 49 | R1, R2, R3 | coincident with Lithoprobe AG-23 |
| Cobalt | 46.375 | R1, R2 | |
| Matheson | 53.95 | R1, R2 | R1, R2 offset |
| Swayze | 89.35, 11.8 | R1, R2 × 3 | |
| Geraldton | 60.2125 | R1 × 2, R2 | Coincident with Lithoprobe WS-3a |
| Sturgeon Lake | 73.475 | R1 | |
| Atikokan | 54.1 | R1 | Crooked line |
| Dryden | 74.4 | R1, R2 × 2 | |
| Rainy River | 33.15 | R1, R2 | |
| Sudbury | 39, 17, 16, 10 | R1 × 3, R2 × 2, R3 | Grid of lines |

## 3. Seismic Data Processing

The Metal Earth seismic surveys cover over 1000 km of the Superior province, stretching from Rainy River in the Wabigoon geological subprovince of westernmost Ontario to Chibougamau in the Abitibi geological subprovince of Quebec. The surveys comprise fifteen R1 regional and fourteen R2 high-resolution surveys (Figure 4; Table 2). Most of the seismic data were acquired along existing paved and gravel roads leading to crooked seismic line geometries. The Metal Earth seismic surveys were processed by Absolute Imaging Inc. and included generating both post-stack and pre-stack migrated seismic sections. Table 3 summarizes the main processing steps and specific parameters that were used for processing the Metal Earth seismic data. Below, we summarize and detail the preliminary processing workflow applied to the Metal Earth seismic data. We also plan to process the R1 seismic data using high-resolution seismic imaging (e.g., pre-stack depth migration) methods.

**Table 3.** Seismic data processing streams used for Metal Earth data.

| Processing Step | Parameters Used | Comment |
|---|---|---|
| Trace Kills and Reversals | | |
| Min Phase Conversion | | |
| Ensemble Balance, Amplitude Recovery | Time power correction + 1.5 | |
| Surface Conistent Scaling | | |
| Linear and Erratic Noise Attenuation | | |
| Surface-Consistent Deconvolution | Operator: 160 ms<br>Prewhitening: 0.1 % | Design window:<br>171–10,000 ms at 38 m offset<br>3347–10,000 ms at 15,000 m offset |
| Anomalous Frequency Suppression | Desired output band: 5–100 Hz<br>Signal band: 15–50 Hz | |
| Refraction Statics | Datum: 500 m<br>Replacement Velocity: 5600 m/s | Tomography |
| Linear and Erratic Noise Attenuation | | |
| TE Mean window | | Design window:<br>171–10,000 ms at 38 m offset<br>3347–10,000 ms at 15,000 m offset |
| Velocity Analysis | | Every 1.0 km |
| Surface Consistent Residual Statics | Max shift 64 ms<br>Window: 2000–9000 ms | |
| Velocity Analysis2 | | Every 500 m |

**Table 3.** *Cont.*

| Processing Step | Parameters Used | Comment |
|---|---|---|
| Surface Consistent Residual Statics | Max shift 48 ms<br>Window: 1000–9000 ms | |
| Post-Stack Time migration | | |
| Velocity Analysis | | |
| Normal Move-out & Mute | | |
| CDP stack | | |
| Time Migration | Kirchhoff Summation | Migration Angle: 65 degrees<br>Max Aperture: 15,000 m |
| Pre-Stack Time migration (PSTM) | | |
| Velocity Analysis (PSTM) | Kirchhoff Summation | VP-CDP Distance: ≤500 m |
| Trace Equalization window | Rolling Window: 1000 ms<br>Overlap 50% | |
| Pre Stack Time Migration (PSTM) | Kirchhoff Summation | Migration Angle: 65 degrees<br>Max Aperture: 10,000 m |
| Front-End Muting | 3/93 1067/758<br>3554/1871 8028/2778 (m/ms) | |
| CDP Stack | | |
| Random Noise Attenuation | | |
| TraceEqualization window | Rolling Window: 1000 ms<br>Overlap 50% | |

*3.1. Geometry Check*

Except for the first three seismic transects (Chibougamau, Malartic, and Rouyn-Noranda), the Metal Earth seismic surveys were acquired with a single sweep per station at 12.5 m and 6.25 m shot spacing for R1 and R2 acquisition scenarios, respectively. This spreading of sweeps for each nominal source point was adapted in order to reduce damage to the roads due to repeated sweeps in the same location. The shots from these surveys were stacked to generate the nominal shot spacing of 50 m and 25 m for R1 and R2 scenarios, respectively. The stacking was applied only for the shot points while the receivers were kept in their original field locations. Figure 3 shows a seismic shot record from the Metal Earth's Malartic R1 survey with reflection signals and various noise types annotated.

Most of the R1 surveys were acquired with live receivers along the entire line. These very long offset recordings can be problematic for Common Mid-Point (CMP) binning of crooked lines, therefore, during processing the maximum offset for R1 surveys was restricted to 15 km. Figure 5a,b show the VP–CMP distance map for Swayze R1 survey before and after restriction to 15 km offset, respectively. A quality control step was used to identify bad records and discard them from further processing. The first breaks were picked for shot records and analyzed by a geometry estimator to detect the geometry errors. After correcting any geometry errors, the binning was applied for crooked lines. The binning for crooked lines sometimes required several iterations to find the optimal binning scenario [16].

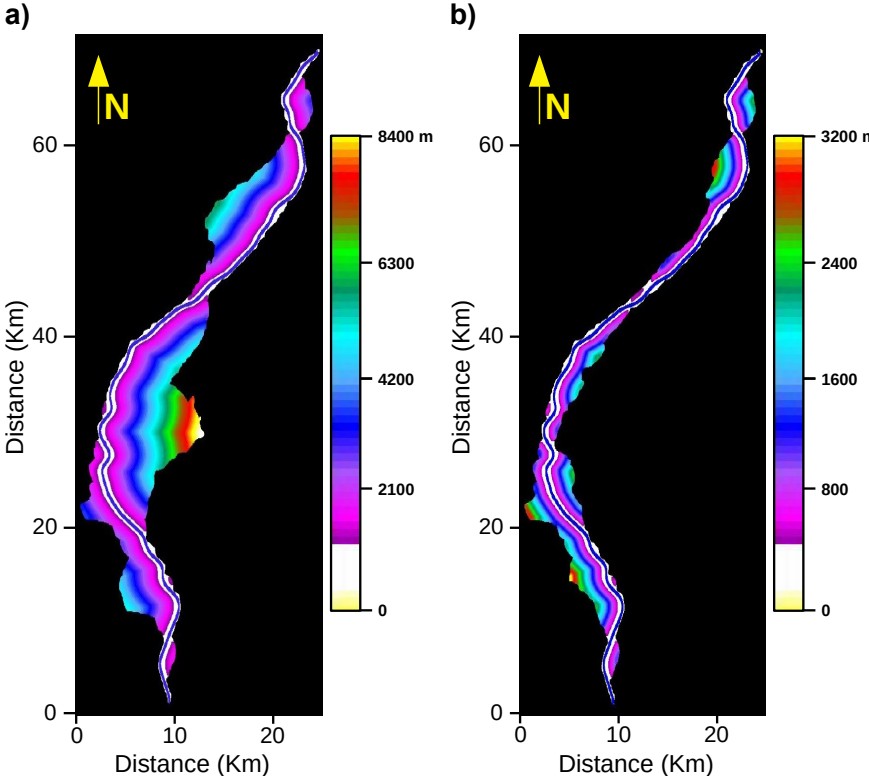

**Figure 5.** VP-CMP distance maps (meters) for Metal Earth Swayze line. (**a**) Unlimited offset. (**b**) Limited offset to 15 km.

### 3.2. Amplitude and Phase Corrections

Balancing the trace amplitudes among the various shots and receivers is necessary for a robust seismic processing work-flow. In order to reduce the variations in amplitude between shot records, an ensemble balance algorithm was used to compute a scalar that balanced all shot record energies. An amplitude recovery with a gain function $g(t) = t^{1.5}$ was applied to compensate the loss of amplitude due to wave-front spreading and attenuation. A trace-equalization algorithm was used after applying each de-noising and deconvolution step during the processing stream. Trace equalization was applied using a single-window, trace-by-trace amplitude balancing algorithm. The seismic records from Vibroseis sources contain zero-phase data after correlation of the Vibroseis source and recorded sweeps. The Vibroseis data were converted to minimum phase records using a minimum phase filter before deconvolution.

### 3.3. Random and Coherent Noise Attenuation

The Metal Earth seismic shot records contained various types of random and coherent noise (Figure 3). Ideally seismic processing flows should effectively attenuate all types of random noise as well as coherent noise such as surface waves, converted waves, and first break arrivals; retaining only reflected wave signals. In order to locally (both in time and space) target and attenuate high-amplitude linear and erratic noise, a time-frequency domain de-noising was applied to the data. Several iterations of harmonic noise suppression were used to attenuate strong 60 Hz signal throughout the data on most of the lines. A frequency domain filtering was applied to suppress any anomalous (non-sweep) frequencies and achieve consistent amplitudes over the entire frequency domain.

### 3.4. Surface Consistent Deconvolution

The recorded seismic signal is the result of the convolution of the source signal with the instruments, the geophones and the response of the crustal rocks. The crustal response includes some undesirable effects, such as reverberation, attenuation and ghost events. The objective of deconvolution is to estimate these effects as linear filters and then design and apply inverse filters to remove them [17–19]. Surface Consistent Deconvolution assumes that a seismic wavelet can be decomposed into its source, receiver, offset and CDP components. Generally, deconvolution performs best when the design window does not include noise such as ground-roll, air blast, and first break reverberations. An operator length of 160 ms and pre-whitening of 0.1% were found to be optimal for Metal Earth seismic data.

### 3.5. Near-Surface Refraction Analysis

Near-surface travel-time irregularities, caused by shallow, low velocity, and unconsolidated weathering layers, can distort the continuity of primary reflections in seismic records. Field static corrections are applied to compensate for the effects of variations in elevation and weathering velocity. The objective is to determine the reflection arrival times which would have been observed if all measurements had been made on a flat plane without the presence of weathered or low-velocity material layers. These corrections are based on the refracted first-break seismic P-waves. The first break picks were used to obtain an initial near-surface model and then a refraction tomography method [20] was used to calculate the near-surface velocity field. A replacement velocity of 5600 m/s and a flat datum of 500 m were used in the final static calculations. Figure 6 shows the near-surface velocity model that was obtained for the Swayze R1 transect.

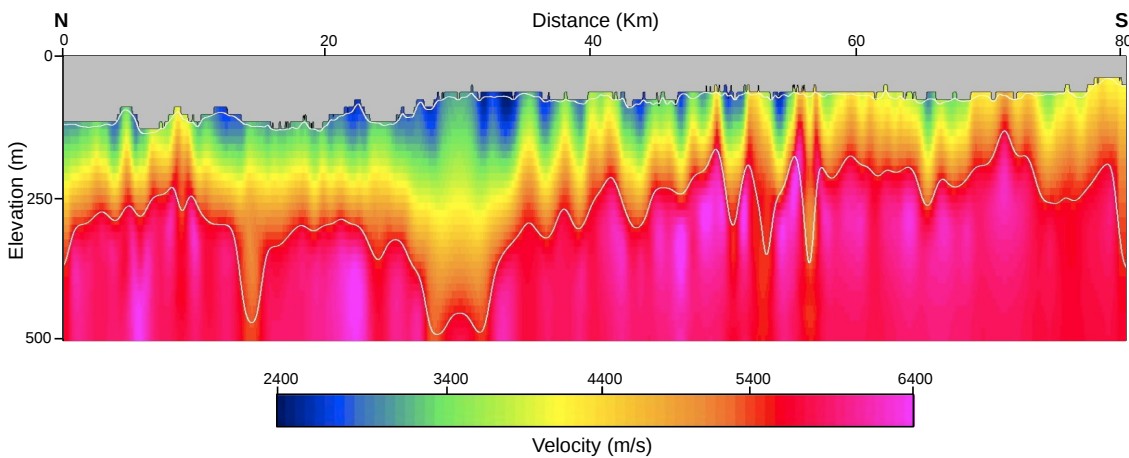

**Figure 6.** Near-surface velocity model for Swayze R1 survey estimated using refraction tomography.

### 3.6. Velocity Analysis and Statics Correction

The velocity analysis used a combination of semblance, super-gather, velocity function stacks, and dynamic stacks. The semblance panel displayed the stack response as a function of time and velocity in a contour plot. Semblances are computed by performing hyperbolic scans at a specified number of constant velocities between a minimum and maximum stacking velocity on the pre-stack data. The gather panel displayed a common offset stacked super-gather of a specified number of CDPs. Normal Move-Out (NMO) correction and time-offset mutes were interactively applied during this analysis. The initial velocities were picked every 1000 m along the transects, while the second and third pass velocities were picked every 500 m.

Refraction field statics removed a significant portion of the long wavelength travel-time irregularities caused by the near-surface weathering layer. Residual static corrections were also required to correct small inaccuracies in the near-surface model. Their application led to a clear

improvement in the final processed section. The residual statics were applied in a surface consistent manner where each trace is cross-correlated with other input traces and time shifts applied to the data are the sum of shot-consistent and geophone-consistent times. The parameters used for residual statics correction are provided in Table 3.

### 3.7. Post-Stack Kirchhoff Migration

The Metal Earth seismic data were first migrated using a Post-Stack Kirchhoff migration method. Before migration, the velocities and mutes finalized at control points and then interpolated across all CDP gathers. The CDP gathers were then NMO-corrected and stacked using an alpha-trimmed mean stack algorithm (10% of highest and lowest samples were excluded from summation) to generate a zero-offset stack. A maximum migration angle of 65 degrees and a maximum aperture of 10,000 m were used for the post-stack Kirchhoff Migration for all of the transects in the Metal Earth project for consistency.

### 3.8. Pre-Stack Kirchhoff Migration

Pre-Stack Time Migration (PSTM) migrates each trace to all output CDP bin centers by accounting exactly for its component surface shot and receiver locations and for any variations in the elevation. Determining the best velocity to use for post-stack time migration is typically based on imaging reflectors that best represent structural geologic features. In contrast, when we run pre-stack time migration velocity analysis of migrated stack panels (and migrated gather panels) we are not only looking at structural fit versus misfit, but also amplitude imaging and energy focusing [21]. In addition, instead of looking at incremental velocity changes of 10% as with the post-stack analysis, we are routinely finding significant structural and imaging changes with only a 2–3% incremental change in velocity. This significant increase in migration velocity precision means we can obtain a more accurate migrated image. The PSTM velocity fields were picked from percentage stack panels and percentage migrated gathers. The migrated stack panels and gathers were generated for 70% to 130% of target velocities (with 2% intervals), and were used for the velocity analysis. The estimated velocity field was extended to gathers using the offset distribution for a given line. The number of output offset planes were dependent on the number of input offset bins for a given line. A maximum migration angle of 65 degrees and a maximum aperture of 10,000 m were used for all lines in the Metal Earth project.

Not all of the data were pre-stack migrated. In some locations, the acquisition geometry was extremely crooked and resulted in very large VP-CMP distances. These large distances would lead to the cancellation of off-line energy during the pre-stack migration process resulting in a less optimal result. Therefore, the VP-CMP distance threshold was limited to 500 m for any pre-stack time migration done in order to mitigate this problem. In Figure 5b, the white region around the CMP line depicts the region in which the CMP distance falls within 0 and 500 m.

### 3.9. Post-Migration Processing

A limited aperture Tau-P transform was applied to the final stack after both post-stack and pre-stack migration, in order to enhance the coherent seismic events. In order to achieve a well-balanced section, a trace-by-trace multi-gate trace-equalization filter was used with a window length of 1000 ms and 50% overlap between the windows. A combination of pre-stack and post-stack noise suppression allowed for a reasonably clean and interpretable seismic section. Figure 7a,b show the Post-Stack and Pre-Stack migrated sections for Metal Earth's Swayze R1 survey, respectively. Here, we have plotted a 30 ms long energy attribute of seismic traces for better visualization of the seismic events. Despite the clear similarities between the sections, pre-stack migration shows higher resolution imaging and separation of dipping reflectors.

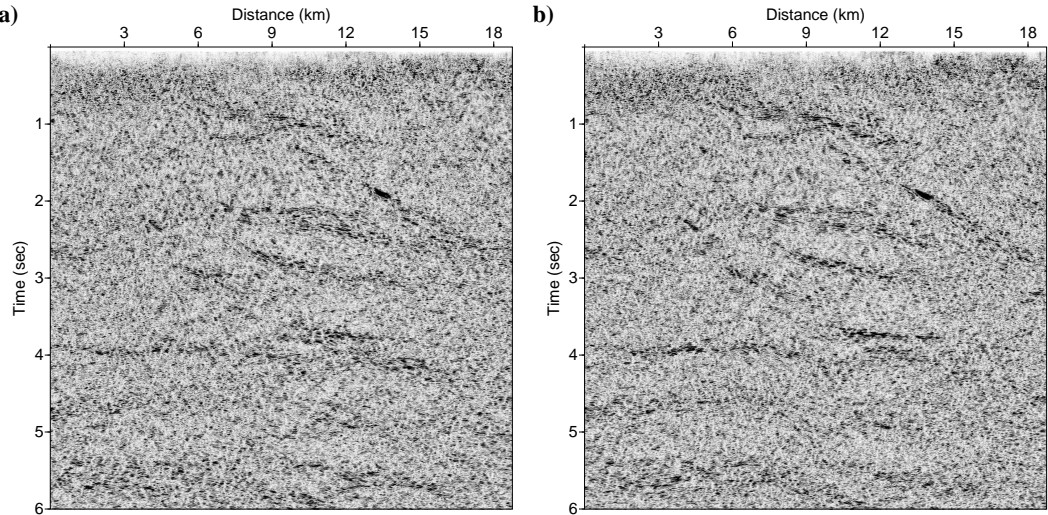

**Figure 7.** Post-Stack (**a**) and Pre-Stack (**b**) migrated sections for the northern part of Swayze R1 survey.

## 4. Discussion

The processing workflow of Metal Earth seismic data focused on robust static solutions, detailed velocity analysis, minimal trace smoothing, and high-resolution imaging. The overall data quality was fairly consistent between various Metal Earth transects. Further high-resolution processing of the seismic data is underway by taking into account the cross-dip corrections and 3D pre-stack imaging of the crooked seismic lines. Some of the Metal Earth seismic transects overlap with some of the Lithoprobe seismic lines (Figure 4). Figure 8a,b show the migrated seismic sections along 32 km of overlapped sections of the Metal Earth's Geraldton R1 and Lithoprobe's WS-3a transects, respectively. Figure 9a,b show the migrated seismic sections along 14 km of overlapped parts of Metal Earth's Larder Lake R1 and Lithoprobe's AG-23 transects, respectively. The overlapped parts of these surveys are marked with green lines (highlighted by green circles) on the map in Figure 4. Metal Earth seismic sections show a distinct improvement in frequency content, dip resolution, and continuity of reflectors in comparison to the Lithoprobe seismic sections in both cases.

It is reassuring that seismic reflection data collected decades apart over the same roads produce very similar reflections but also that the more recent version is a clear improvement in resolution and lateral continuity which are important for the geologic interpretation of the reflections. The 2017 Metal Earth Larder Lake transect coincided with a portion of the 1990 Lithoprobe Abitibi-Grenville line 23 transect along Ontario Provincial highway 624 just south of the town of Larder Lake (Figures 4 and 9). A comparison of the acquisition specifications used (Table 4) indicates a significantly wider bandwidth was used by Metal Earth. Metal Earth also used significantly shorter intervals between geophones and vibrator points and did not use strings of geophones or form vibrator arrays. The 2017 survey was done in fine autumn weather whereas the 1990 acquisition occurred after a heavy snowstorm and during rain with icy roads (average temperatures from −4 to 0 °C). The comparison of similarly processed seismic sections reveals smeared 'blobs' of reflectivity on the AG23 section where the newer Larder Lake section resolves listric curves nearly twice the length (Figure 9). Areas with apparently no reflectivity on AG23 now have many layers of reflectors. Significantly, the maximum depth of these laterally continuous shallowly dipping reflections is 12 s (about 38 km) on the Larder Lake section, but only 9 s (about 30 km) on AG23. Because this deepest zone of reflectors is commonly assumed to represent the Moho, such a large discrepancy is significant. Independent determinations of Moho depth near Kirkland Lake indicate a depth of 38 km [22].

**Table 4.** Comparison of acquisition parameters used in Metal Earth and Lithoprobe surveys.

| Parameter | Metal Earth (R1) | Lithoprobe WS 3a | Lithoprobe AG 23 |
|---|---|---|---|
| Year | autumn, 2017 | March, 2000 | November, 1990 |
| Contractor | SAExploration | Kinetex | J.R.S. Exploration |
| Record length | 12 or 16 s | 18 s | 18 s |
| Sample rate | 2 ms | 4 ms | 4 ms |
| Spread size | 15 km–0–15 km | 12 km–0–12 km | 8.1 km–0–8.1 km |
| Vibrator; length | I/O AHV-IV 364; 10 m | I/O AHV-IV 360; 10 m | Mertz-18; 11 m |
| Source interval | 12.5 m (1 sweep) 50 m (4 sweeps 0 or 1 m move up) | 100 m (8 sweeps) 7.14 m move up | 100 m (8 sweeps) 7.14 m move up |
| Vibrator sweep | 28 s, 2–96 Hz linear; 4 vibs | 28 s, 10–84 Hz linear; 4 vibs; | 14 s, 10–56 Hz linear; 4 vibes |
| Receiver interval | 25 m | 25 m | 50 m |
| Geophones | 5 Hz; Single | 10 Hz; 9 over 25 m | 14 Hz; 9 over 50 m |

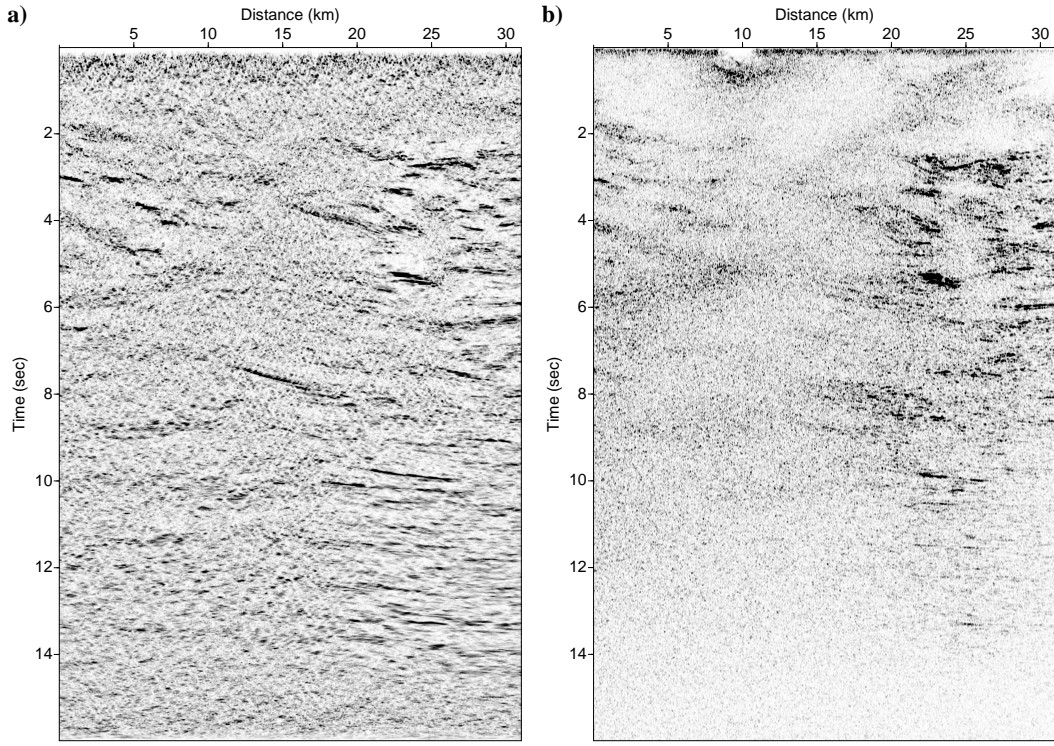

**Figure 8.** Comparison betweem the overlapped portions of Metal Earth's Geraldton R1 Pre-stack migrated section (**a**) and Lithoprobe's WS-3a post-stack migrated section (**b**).

A similar comparison is possible in the Wabigoon greenstone belt near the town of Geraldton, Ontario where the 2017 Metal Earth Geraldton transect was acquired along the same gravel roads as the 2000 Lithoprobe Western Superior Line 3a (Figures 4 and 8). The Metal Earth transect was done during the first snowfalls whereas the Lithoprobe line was acquired after a heavy April snowfall and daily melting of the surface created poor geophone coupling. Similar to the Larder Lake example, reflections are more numerous, more laterally continuous and better resolved on the Metal Earth section (Figure 8). Apparently non-reflective parts of the Lithoprobe section are now revealed to have shallowly dipping reflectors. Here well-defined reflectors occur as deep as 14 s (about 45 km) on the Metal Earth section but only as deep as 10 s (about 32 km) on the Lithoprobe section.

The increased vertical resolution and definition of several reflectors whereas previously only one fuzzy reflection was observed can be attributed to the increased bandwidth available to the processing of the more recent survey. Lateral continuity of individual reflectors is undoubtedly increased by the higher spatial resolution of vibrator, geophone and common depth points available in the latter

survey. This advantage was further augmented by the use of pre-stack migration. Utilizing Full Waveform Inversion (FWI) and Pre-Stack Depth Migration (PSDM) methods could lead to even higher resolution subsurface images. Lithoprobe transects concentrated their effort in both acquisition and processing of the data on the lower crust and upper mantle whereas Metal Earth regional transects focused on the upper and middle crust although both strategies sought quality whole-crust seismic sections in general.

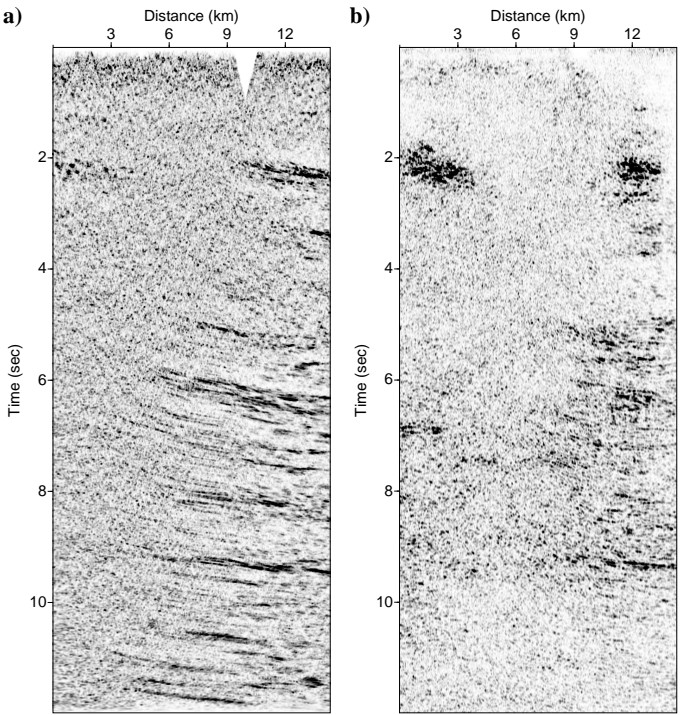

**Figure 9.** Comparison betweem the overlapped portions of Metal Earth's Larder Lake R1 Pre-stack migrated section (**a**) and Lithoprobe's AG-23 post-stack migrated section (**b**).

## 5. Conclusions

The 2017 Metal Earth seismic surveys comprised 15 R1 (regional) and 14 R2 (high-resolution) surveys. The survey's 927 line kilometers of profiles cover Archean Canada from Rainy River near the Manitoba-Ontario border in the Wabigoon geological subprovince to Chibougamau in eastern Quebec in the Abitibi geological subprovince. Use of broader bandwidth vibrator sources and geophones improved potential resolution of reflectors at depth. More spatially compact source and receiver arrays, spaced at shorter intervals than in previous Lithoprobe surveys, improved lateral reflector definition and resolution. The processing workflow for the Metal Earth seismic data focused on robust static solutions, detailed velocity analysis, minimal trace smoothing, and high-resolution migrations using wide apertures. The overall data quality was good and fairly consistent among the thirteen Metal Earth transects. These reflection seismic data will play a central and crucial role in understanding the tectonic and geological differences between the Abitibi (metal-endowed) and Wabigoon (less-endowed) subprovinces.

**Author Contributions:** Seismic Data Acquisition and Design, D.S., J.M., M.N., and S.C. (Saeid Cheraghi); Seismic Data Processing, M.N., S.F., S.C. (Samo Cilensek ), E.F., and S.C. (Saeid Cheraghi); writing—original draft preparation, D.S. and M.N.; writing—review and editing, M.N. and D.S.

**Funding:** This research was funded by the Canada First Research Excellence Fund (CFREF).

**Acknowledgments:** We thank SAExploration for acquiring and Absolute Imaging Inc. for processing the seismic data.

**Conflicts of Interest:** The authors declare no conflict of interest.

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
