# Peer review of "Acquisition and Processing of Wider Bandwidth Seismic Data in Crystalline Crust: Progress with the Metal Earth Project"

_minerals, doi:10.3390/min9030145_

Round 1
Reviewer 1 Report
See separate report

Author Response
General
I have no trouble in recommending this paper for publication in the special issue with minor
corrections. On the it is clear and well-written.
It describes the acquisition and processing of a large amount of data which, in earth science
research terms, must have been very expensive. I am impressed by the data quality, particularly
with the comparisons of vintage crustal reflection data from 20-30 years ago when I was active in
the field. I look forward to seeing geological interpretations in print in due course.
My review should be transmitted to the authors WITH my name on it (I have made that a condition
of reviewing since about 1975, as I am against anonymous refereeing).
Answer) Thank you for reviewing our paper and providing valuable comments. We were also impressed when saw the good match between Lithoprobe and Metal Earth surveys in overlapped areas. We have this paper just to give a general overview of the Metal Earth seismic surveys. Indeed there are more processing work and interpretation is underway and we will have them in print as they get ready.
Grammar and typos
Title: broader bandwidth not 'wider because 'broad bandwidth' is the normal technical usage.
'Wider' sounds a bit amateur.
Answer) Thanks for pointing this out. We used broader instead of wider in the title.
Abstract
"rated down to 5Hz frequencies" is then followed by "2 Hz signal" - sounds inconsistent. I suggest
change the whole sentence to:
"Vibroseis acquisition with receivers having a 5 Hz response (10 dB down) generated records from
a sweep signal starting at 2 Hz, sweeping up to 150 or 200 Hz"
A) Thanks for suggesting this as well. We do like your revised sentence and used it in the text.
Text
line 68. What are "creative" vibrator move-ups? Need to explain what this is.
A) we used the term “spatially-spread” vibrator move-ups. We explain this later and also in the acquisition table. The Vibrators usually sweep at a single shot station for 4 times with zero move-ups to generate the stacked shot gathers. Doing so could lead to damage to paved roads, therefore we used spatially-spread move-ups between shots points. This means that the shots form R1 surveys, that was supposed to be 50m apart (by sweeping 4 times at each shot point), was gathered by doing a single sweep at every 12.5 m interval.
Figure 2. You can't really claim that the bandwidth in the left-hand image is 2-6 because the
amplitude at 2 Hz is -20 dB. It is more honest to say '3-6 Hz' and explain that 3 Hz is the -10 dB
point. That means the image is effectively one octave. Similarly, change the label on the central
image to 3-16 Hz.
A) Yes. These are the allowed band-pass frequencies and as you mentioned the 2Hz is -20 dB. However sometimes seismic wave mode conversions can generate low-frequency energies. Since these plots are indeed band-passed in the ranges indicated by limits on the plots, we left the plots as they were. We added to the caption that these are the band-pass frequencies allowed. We understand your point that most likely band-pass 3-6 Hz will look the same as 2-6Hz for the signal content.
Lines 88 and 97. Figure 4 is mentioned Figure 3. You need to either swap around some text,
or else renumber the figures.
A) We changed the order of Figures and renumbered them. Thanks for noticing this.
Table 3. This needs to be recast because the rows are all out of alignment. Could be done by putting
borders all the cells.
A) we used dashed-lines to separate the rows.
Table 3. Anomalous frequency suppression; 'Outband' is new to me. Do you mean Reject band?
A) We mean “Desired output band”. We changed “” to “desired output band” in Table 3.
Lines 224-5 "... with the output going directly to stack" isn't clear to me, PSTM (to me)
implies that time migration is applied, followed by .
A) We meant when the velocity analysis finished, the PSTM applied and then stacked. You are right that sentence was confusing and we removed that sentence.
Line 227. "very large CMP distances" is ambiguous. I presume you mean 'very large VP-CMP
offsets'
A) Thanks for catching this typo as well. We meant VP-CMP distance and we made the correction.
Figure 9. There are impressive reflectors down to 12 s, taken to be the depth. It would be
better to extend the length of the panel to 13 or 14 s, to show that below the moho there are no
more reflectors.
A) Unfortunately, the Metal Earth’s Lake R1 correlated traces were only preserved till 12 seconds and uncorrelated recording are no longer available to recover time sample beyond 12 seconds. The Geraldton survey, shown in Figure 8, has 16 seconds record length and the Moho can be seen clearly in that survey.
line 70 should be "a few depth …"
Lines 139- 140. Should be "very long-offset recordings" (plural)
Line 143, "... and discard ..." (infinitive tense; not 'discarded')
Line 222 Dependent, not .
Line 263 'was done in fine …'
Line 277 'more laterally continuous'
Line 285 'leverage' should be past tense. But its an ugly verb, coming from the business world
(ratio of loans to equity, or something like that). Why not simple English like 'helped' or
'augmented
A) All of the above typos are corrected. Thanks again for your thorough reading of our manuscript.
Reviewer 2 Report
Dear Authors,
Thank you very much for your research. I enjoyed reading the paper. The paper is nicely written and the results and the acquired data especially look very interesting to me. I do recommend the paper for publication and suggest a moderate revision. Below are my comments, which I strongly believe will improve the paper.
First, there is a major inconsistency in the narrative. In the abstract and in the section ``2.3’’ you are speculating about FWI. While reading the paper I was waiting for the corresponding results but have not seen them. I propose to highlight the main message of the manuscript: excellent acquisition and preliminary processing results – and refrain from elaborate but not necessary discussion about FWI. Of course, I would mention it in the discussion section and in the future plans but would not leave it next to the descriptions of the current results of the paper. Could you please revise the manuscript in this light?
Second, I strongly suggest including depth migration in the discussion. Apparently, the area is of high geological complexity, and it would be great to see depth migration results. Refer to the literature – vast improvement can be achieved. Please elaborate. This could be the intermediate step between your current time-domain processing stage and FWI.
Third, I think the paper would strongly benefit from including the figures of your flattened time-migration gathers. You discuss them but never include them. Please do so.
Fourth, I think mentioning acquisition parameters is important but you do it several times in different places for different transects, and I got lost. Please make it more consistent.
Fifth, you only show results from R1 profile. I would include some analysis of other profiles. Even preliminary. Please elaborate what is the status on them. Otherwise, they seem to be lost while reader moves to conclusions.
Minor suggestions:
Page 2 line 42: Although sounds good I would not call earthquakes and explosions ``White Noise’’. This is a pretty strong signal, which is indeed coherent and thus is not white.
Page 2 line 55: I have not encountered the acquisition aspect of geophyscis for a while so may be it is OK. Is ``move up’’ a term? I have not seen it. Please change it if appropriate.
Page 2 line 60: what is a ``natural’’ frequency of a geophone?
Page 2 line 61: ``the recorded signal is down only a few’’: are you referring to higher frequency panels in Figure 2? Please clarify.
Page 2 line 68: what are ``creative vibrator move-ups’’?
Page 4 line 115: you want to use 80 km offsets then on page 6 line 141 you cut them at 15 km. This is inconsistent. May be focus on the processing steps for R1. Otherwise, all the scales get tangled. Please revise and make consistent: what you have done for R1, what processing you already have or will have for R2 and R3.
Page 4 line 81: not sure how longer source intervals would improve spatial resolution.
Table 3: you talk about lower frequencies in acquisition but then cut the signal from 15-50 Hz. This is inconsistent.
Page 6 line 136: what do you mean by ``The stacking was applied with shot point and receivers kept in their original field locations’’?
Please provide citations for surface-consistent decon.
I got confused by the section 3.7. How do you obtain gathers after post-stack Kirchhoff migration? How do you NMO-correct them? Migration stacks all the offset information.
Page 10 line 221: What do you mean by ``velocity field was migrated to gathers’’?
Page 11 line 263: ``is fine’’ -> ``in fine’’
Thank you very much for your consideration. Looking forward to the revised version of the paper.
Author Response
Thank you very much for your research. I enjoyed reading the paper. The paper is nicely written and the results and the acquired data especially look very interesting to me. I do recommend the paper for publication and suggest a moderate revision. Below are my comments, which I strongly believe will improve the paper.
Answer) Thank you for reviewing our Article and providing very useful comments. Please find below our answers (A) to your comments.
First, there is a major inconsistency in the narrative. In the abstract and in the section ``2.3’’ you are speculating about FWI. While reading the paper I was waiting for the corresponding results but have not seen them. I propose to highlight the main message of the manuscript: excellent acquisition and preliminary processing results – and refrain from elaborate but not necessary discussion about FWI. Of course, I would mention it in the discussion section and in the future plans but would not leave it next to the descriptions of the current results of the paper. Could you please revise the manuscript in this light?
A) The emphasis in this subsection is about the acquisition parameters that will be suitable for FWI imaging of acquired seismic data. We add a sentence to make it clear that we will not show any FWI analysis and results on this paper the end of Introduction section: “In this article, we will only discuss the seismic processing workflow for the crustal scale (R1) seismic data. The processing and analysis of high-resolution (R2) and full waveform inversion (R3) seismic data is currently underway and will be discussed in future publications.”
Second, I strongly suggest including depth migration in the discussion. Apparently, the area is of high geological complexity, and it would be great to see depth migration results. Refer to the literature – vast improvement can be achieved. Please elaborate. This could be the intermediate step between your current time-domain processing stage and FWI.
A) Yes. The geological setting in study areas are very complex and indeed depth migration can improve the seismic images. However, finding a reliable interval velocity for the depth migration can be challenging. We added a line in discussion to emphasis the importance of Depth migration: “Utilizing Full Waveform Inversion (FWI) and Depth Migration methods could lead to even more accurate subsurface seismic images.”
Third, I think the paper would strongly benefit from including the figures of your flattened time-migration gathers. You discuss them but never include them. Please do so.
A) We will be discussing flattened PSTM gathers in a more specialized processing publication. In this paper, we tried to keep the processing details to a minimum and try to focus on the general approach and show a few final migrated sections and compare with previously done surveys in study area such as Lithoprobe data. We prefer to expand more on processing intermediate steps and QCs in a more processing specialized paper.
Fourth, I think mentioning acquisition parameters is important but you do it several times in different places for different transects, and I got lost. Please make it more consistent.
A) We have to repeat some acquisition parameter to clarify the processing steps in few parts. Also, there are minor differences between various transects that we had to clarify these. We made sure there is no inconsistencies in our description of acquisition parameters in the revised manuscript.
Fifth, you only show results from R1 profile. I would include some analysis of other profiles. Even preliminary. Please elaborate what is the status on them. Otherwise, they seem to be lost while reader moves to conclusions.
A) We added a line at the end of introduction section stating that in this paper we will only discuss the processing of R1 surveys and the processing and results from R2 and R3 surveys are underway and to be reported in future as they get completed.
Minor suggestions:
Page 2 line 42: Although sounds good I would not call earthquakes and explosions ``White Noise’’. This is a pretty strong signal, which is indeed coherent and thus is not white.
A) We agree. Thanks for pointing it out. We replaced “White Noise spectral sources” with “The global seismology sources”.
Page 2 line 55: I have not encountered the acquisition aspect of geophyscis for a while so may be it is OK. Is ``move up’’ a term? I have not seen it. Please change it if appropriate.
A) Move-up is often used in Vibroseis acquisition meaning how much vibroseises move spatially between multiple sweeps in a shots stations.
Page 2 line 60: what is a ``natural’’ frequency of a geophone?
A) The Natural Frequency is the resonant frequency of the spring-mass system in the working axis of the geophone.
Page 2 line 61: ``the recorded signal is down only a few’’: are you referring to higher frequency panels in Figure 2? Please clarify.
A) No. We are referring to the low frequencies around 3 Hz. We stated 3Hz at the end of sentences.
Page 2 line 68: what are ``creative vibrator move-ups’’?
A) we used the term “spatially-spread” vibrator move-ups. We explain this later and also in the acquisition table. The Vibrators usually sweep at a single shot station for 4 times with zero move-ups to generate the stacked shot gathers. Doing so could lead to damage to paved roads, therefore we used spatially-spread move-ups between shots points. This means that the shots form R1 surveys, that was supposed to be 50m apart (by sweeping 4 times at each shot point), was gathered by doing a single sweep at every 12.5 m interval.
Page 4 line 115: you want to use 80 km offsets then on page 6 line 141 you cut them at 15 km. This is inconsistent. May be focus on the processing steps for R1. Otherwise, all the scales get tangled. Please revise and make consistent: what you have done for R1, what processing you already have or will have for R2 and R3.
A) We hope that by adding the two sentences at the end of the Introduction section, it will be clear that we are only talking about processing R1 surveys in this paper. Elaborating on R2 and R3 processing steps will make thing even more confusing in this paper and we will leave it for future publications.
Page 4 line 81: not sure how longer source intervals would improve spatial resolution.
A) Here we mean larger interval between sweeps for a single shot point. The Vibrators usually sweep at a single shot station for 4 times with zero move-ups to generate the stacked shot gathers. Doing so could lead to damage to paved roads, therefore we used spatially-spread move-ups between shots points. This means that the shots form R1 surveys, that was supposed to be 50m apart (by sweeping 4 time at each shot point), was gathered by doing a single sweep at every 12.5 m interval.
Table 3: you talk about lower frequencies in acquisition but then cut the signal from 15-50 Hz. This is inconsistent.
A) That is the strong signal band used to remove anomalous Frequencies from the desired output band of 5-100 Hz. We do not cut the signal. We changed “outband” to “desired output band” in table 3.
Page 6 line 136: what do you mean by ``The stacking was applied with shot point and receivers kept in their original field locations’’?
A) The current processing of R1 surveys was performed by stacking 4 equally-spaced sweeps between shots points into one stacked shot. However, the receiver are kept in their original configuration. We modified the sentence to this: “The stacking was applied only for the shot points while the receivers were kept in their original field locations.”
Please provide citations for surface-consistent decon.
A) We added “Cary and Lorentz, 1993” to the references for Surface consistent deconvolution.
I got confused by the section 3.7. How do you obtain gathers after post-stack Kirchhoff migration? How do you NMO-correct them? Migration stacks all the offset information.
A). Sorry for the confusion that was mainly due to the structure of the sentence. we meant CDP gathers and velocities before starting post-stack migration. We made corrections to the sentence as below: “The Metal Earth seismic data were first migrated using a Post-Stack Kirchhoff migration method. Before migration, the velocities and mutes finalized at control points and then interpolated across all CDP gathers.”
Page 10 line 221: What do you mean by ``velocity field was migrated to gathers’’?
A) We mean it was extended to gathers. We changed “migrated” to “extended”.
Page 11 line 263: ``is fine’’ -> ``in fine’’
A) Thanks for noticing this. It was corrected.
Thank you very much for your consideration. Looking forward to the revised version of the paper.
22 January 2019
29 Jan 2019 02:26:13

Round 2
Reviewer 2 Report
Dear Authors,
I thinks your edits clarified and addressed my concerns. I recommend the paper for publication.
Please still consider the following:
for surface consistent decon please add https://library.seg.org/doi/abs/10.1190/1.1441133. This is the classical one.
I would still rename the section ``using full waveforms'' to ``future plans on using full waveforms''.
After you get FWI velocities you can try them for depth migration.
Thank you for your research and good luck!
Author Response
I thinks your edits clarified and addressed my concerns. I recommend the paper for publication.
Please still consider the following:
for surface consistent decon please add https://library.seg.org/doi/abs/10.1190/1.1441133. This is the classical one.
A) Thanks for suggesting this reference. We added this reference to our article.
I would still rename the section ``using full waveforms'' to ``future plans on using full waveforms''. After you get FWI velocities you can try them for depth migration.
A) We changed the title of subsection to “Toward using Full waveforms”. We indeed are looking forward for ways of improving the velocity models to be able to do Depth migration.
Thank you for your research and good luck!
p { margin-bottom: 0.1in; line-height: 115%; }